# On Advances of Lattice-Based Cryptographic Schemes and Their Implementations

Harshana Bandara [1] , Yasitha Herath [1], Thushara Weerasundara [1] and Janaka Alawatugoda [2,3,*]

1 Department of Computer Engineering, University of Peradeniya, Peradeniya 20400, Sri Lanka
2 Research & Innovation Division, Faculty of Resilience, Rabdan Academy,
  Abu Dhabi P.O. Box 114646, United Arab Emirates
3 Institute for Integrated and Intelligent Systems, Griffith University, Brisbane, QLD 4111, Australia
* Correspondence: jalawatugoda@ra.ac.ae

**Abstract:** Lattice-based cryptography is centered around the hardness of problems on lattices. A lattice is a grid of points that stretches to infinity. With the development of quantum computers, existing cryptographic schemes are at risk because the underlying mathematical problems can, in theory, be easily solved by quantum computers. Since lattice-based mathematical problems are hard to be solved even by quantum computers, lattice-based cryptography is a promising foundation for future cryptographic schemes. In this paper, we focus on lattice-based public-key encryption schemes. This survey presents the current status of the lattice-based public-key encryption schemes and discusses the existing implementations. Our main focus is the learning with errors problem (LWE problem) and its implementations. In this paper, the plain lattice implementations and variants with special algebraic structures such as ring-based variants are discussed. Additionally, we describe a class of lattice-based functions called lattice trapdoors and their applications.

**Keywords:** post-quantum cryptography; lattice-based cryptography; LWE problem; ring-LWE; lattice trapdoors; implementation

## 1. Introduction

The majority of present-day internet communication utilizes public-key encryption. Public-key encryption is a method that uses two keys to encrypt data. There are two keys known as the public key and the private key. The public key is known to anyone. Data are encrypted using a public key and can only be decrypted by using a private key. This is also known as asymmetric encryption because different keys are used to encrypt and decrypt. Public-key encryption schemes mainly consist of three algorithms, i.e., key generation algorithm, encryption algorithm, and decryption algorithm. Presently, the RSA algorithm [1] and the Diffie–Hellman protocol [2] are used on the Internet as public-key cryptography methods to provide confidentiality for data. Internet protocols such as transport layer security (TLS) and secure sockets layer (SSL) apply these two cryptographic schemes, and, therefore, web/mobile applications, email, messaging, VoIP, WAN, and LAN network communications use these public encryption schemes. For a couple of decades, these systems were believed to be reasonably secure against attacks.

With the development of science and technology, large-scale quantum computers will be available in the near future. Quantum computers are devices that use the properties of quantum physics to store data and perform computations. This can be very useful for certain tasks where they could outperform even the best supercomputers. In a couple of decades, large-scale quantum computers will efficiently solve the hard mathematical problems on which the aforementioned public-key cryptography methods are based. Thus, quantum-safe cryptosystems will be essential for secure communication in the future. Lattice-based cryptography is one of the most promising quantum-safe cryptography candidates, which offers protection against quantum computers.

*Organization of the Paper*

In this paper, we will discuss lattice-based quantum-safe cryptographic schemes. In Section 2, the state of the present-day classical public-key encryption schemes against quantum attacks is discussed. Moreover, a summary of post-quantum cryptographic schemes is presented alongside an introduction to lattice-based cryptographic schemes. In Section 3, different learning with errors (LWE) approaches based on plain lattices and ring lattices are discussed and compared. Section 4 discusses details about the ring-LWE problem, while Section 5 discusses using lattice-based cryptographic schemes as trapdoor functions and their applications. Section 6 presents a summary of cryptographic implementations that employ the LWE problem and variants of the LWE problem.

## 2. Background

In this section, we discuss the widely used non-quantum-safe public-key cryptographic schemes and quantum-safe approaches in cryptography. We give an emphasis on lattice-based cryptography in this section.

### 2.1. Non-Quantum-Safe Public-Key Cryptography

In this section, we give a brief overview of the RSA encryption scheme and the Diffie–Hellman key exchange protocol.

#### 2.1.1. RSA algorithm

This is an asymmetric encryption method built on the fact that it is hard to factorize a large integer into its two prime factors. There are two main parts of the RSA algorithm.

- **Key generation**—this is concerned with generating the public/private keys of the algorithm. $(e, n)$ is the pair of positive integers that is the public key. $(d, n)$ is the pair of positive integers that is the private key. $n$ is computed as the product of two prime numbers $p$ and $q$:

$$n = p \times q$$

  $p$ and $q$ are large prime numbers that are chosen at random. Only $n$ is the publicly known value, and $p$ and $q$ are hidden from the public because factorizing $n$ into $p$ and $q$ is hard. Then, the integer $d$ is picked at random such that it is relatively prime to $p - 1$ and $q - 1$. The integer $e$ is computed from $p$, $q$, and $d$ to be the multiplicative inverse of $d$. Then we have the following.

$$e \times d \equiv 1 \mod (p-1)(q-1)$$

- **Encryption/Decryption algorithms**—this is concerned with the steps to encrypt and decipher the data. First, the message is represented as an integer between 0 and $n - 1$. The encryption is done by raising the message $M$ to its $e$-th power modulo $n$ to obtain the ciphertext $C$. To decrypt the ciphertext $C$, raise it to the power $d$ and modulo $n$. The encryption algorithm is denoted by Enc and the decryption algorithm is denoted by Dec.

$$\text{Enc}(M) = M^e \mod n$$

$$\text{Dec}(C) = C^d \mod n$$

The RSA algorithm can be used for encrypting data as well as generating digital signatures. Since prior key sharing is not needed, there is no need for a secure connection to transfer the secret key. However, factorizing $n$ to $p$ and $q$ will not be hard if quantum computers are available, and, therefore, in the future, the security of this scheme is at risk.

2.1.2. Diffie–Hellman Key Exchange Protocol

The Diffie–Hellman key exchange protocol came into being in 1976. This can be (primarily) used to establish a mutual secret between two parties (Alice and Bob). In this protocol, Alice picks a random integer $a$ and computes $A \leftarrow g^a$ and sends it to Bob over the Internet. Upon receipt of $A$, Bob picks $b$ at random, computes $B \leftarrow g^b$, and sends it to Alice over the Internet. Note that $g$ is the generator of the group which is a priori agreed on by both Alice and Bob. Finally, Alice computes $K \leftarrow B^a$ and Bob computes $K \leftarrow A^b$. In this case, without revealing $a$ or $b$, Alice and Bob can compute a shared secret $K$. In order to compute $K$, the eavesdropper needs to extract either $a$ or $b$, which is exactly solving the discrete logarithm problem. When considering the discrete logarithm problem it means for given $y$, finding $x$ with $y = g^x \mod p$, where $p$ is a large prime and $g$ is a generator as above mentioned is known as a hard problem. Therefore, the eavesdropper cannot find $a$ or $b$ easily.

In 1994, Peter Shor introduced an algorithm for quantum computers [3], which enables an attacker with a sufficiently large quantum computer to solve integer factorization problems and discrete logarithm problems in polynomial time. Since the RSA scheme [1] depends on the hardness of integer factorization and the Diffie–Hellman protocol [2] depends on the hardness of solving the discrete logarithm problem, the security of these public-key encryption schemes are at risk when facing with a quantum computer. Therefore, scientists are currently working on ways to protect against potential quantum-based attackers in near future. In this survey, we will be focusing on the learning with errors (LWE) problem, which is derived from lattice-based cryptography because in the future when quantum computers come to day-to-day use, the Diffie–Hellman key exchange protocol and the RSA algorithm will no longer can provide the required security level.

*2.2. Quantum-Safe Cryptography*

Cryptographers have already made significant strides toward designing quantum-safe cryptographic schemes. Apart from lattice-based cryptography, there are multiple approaches that are believed to be quantum-safe. A couple of the most prominent approaches are briefly discussed in this section.

2.2.1. Hash-Based Cryptography

A signature scheme was built upon the ideas of Lamport's one-time signature (OTS) scheme [4]. In 1979, Merkle [5] introduced a public-key signature scheme based on OTS. Since these schemes produce relatively larger digital signatures, they are not used for real-world applications. Recently, with the need for post-quantum cryptographic schemes, new work on hash-based cryptography has started.

One-time signature blocks that sign a single message per key are used as the building block for hash-based cryptography. The security of this is based on the security of hash functions. Merkle improved this into an algorithm, called SHA-2, that could be used multiple times. Currently, the SHA-2 algorithm is believed to be quantum-safe and hash-based signature schemes that use SHA-2 can be used for post-quantum cryptographic schemes. Apart from security, smaller private/public keys are another advantage of this scheme.

2.2.2. Code-Based Cryptography

Code-based cryptography focuses on cryptographic approaches that use error-correcting codes. Seminal work on this was done by McEliece [6] and Niederreiter [7]. These schemes can be used for encryption, hashing, and signature generation with reasonable efficiency. One of the drawbacks of the original scheme proposed by McEliece is the larger public key size. Since this scheme has shown no vulnerability to known quantum-based attacks, research in this field is regarded as one of the main approaches for quantum-safe cryptography.

### 2.2.3. Multivariate-Quadratic-Equations Cryptography

By using the public-key encryption scheme presented by Matsumoto and Imai [8], which is based on multivariate polynomials of degree two over a finite field, Patarin [9] implemented a cryptosystem. This cryptosystem enjoys computationally efficient and fast operations, which makes the system suitable for deployment on smaller devices. However, the large key size is a major drawback of this system. The security of this system is based on the hardness of solving systems of multivariate equations. Since it is proven to be an NP-complete problem, this scheme is a promising post-quantum cryptographic scheme for signature generation.

### 2.2.4. Secret-Key Cryptography

The best example for this is the symmetric-key encryption scheme based on the AES cipher by Daemen and Rijmen [10]. Grover's algorithm is one of the quantum algorithms that can be used to perform an exhaustive key-search attack on AES [11]. This can produce an output with $O(\sqrt{n})$ evaluations of the function when $n$ is the size of the function's domain. Using 256-bit keys has been proven to be reasonably secure against this attack.

### *2.3. Lattice-Based Cryptography*

Lattice-based cryptography is a promising approach to quantum-safe cryptography. Here we discuss details about lattice-based cryptography.

### 2.3.1. Lattices

For a better understanding of this survey, basic knowledge of lattices is needed. First, consider the mathematical representation of a vector space,

$$\text{span}(B) = \left\{ \sum a_i b_i : a_i \in \mathbb{R} \right\}$$

A vector space is represented by a combination of any arbitrary real coefficients with its basis. A lattice $\mathbb{L}$ is a set of points that spreads to infinity and can be represented by linear combinations of a vector called basis $B = \{b_1, b_2, \ldots, b_n\}$. Therefore, a lattice can be mathematically represented by,

$$\mathbb{L}(B) = \left\{ \sum a_i b_i : a_i \in \mathbb{Z} \right\}$$

The difference between a lattice and a vector space is that, while the basis of a vector space can be combined with any real coefficients, in a lattice only integers are allowed in combination with the basis. This results in lattices having a discrete set of points in space. Since $b_1, \ldots, b_n$ are linearly independent, any point $p$ in the vector space can be represented in a unique combination of $B$ as $p = a_1 b_1 + \cdots + a_n b_n \in \text{span}(B)$. Then, $p \in \mathbb{L}(B)$ if and only if $a_1, \ldots, a_n \in \mathbb{Z}$. Since $B$ is a basis of $\mathbb{L}(B)$, it is also a basis for $\text{span}(B)$ as well. However, not every basis of $\text{span}(B)$ becomes a basis for $\mathbb{L}(B)$.

This definition for lattices can be represented in matrices with linearly independent columns. For decades cryptographers have researched hard problems based on lattices using matrix representations. In cryptography, lattices with a higher number of dimensions are considered. The representation of a 3-D lattice is shown in Figure 1.

In 1996, Ajtai [12] found a reduction for hard lattice problems to average-case from worst-case. This gave a security proof for cryptographic approaches which rely on hard lattice problems. Using these findings, cryptographic schemes based on lattices can be implemented and proven to be secure, as long as the underlying lattice problems are hard to solve in the worst-case scenario.

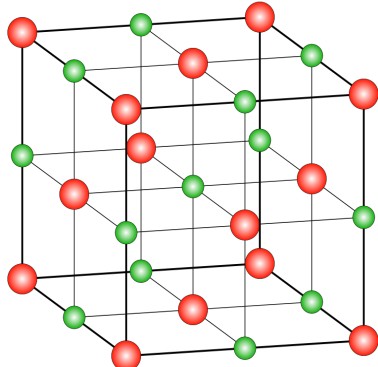

**Figure 1.** Representation of a 3-D lattice (https://www.pngwing.com/en/free-png-yywbt), accessed on 11 September 2022.

2.3.2. Ajtai's Work

Notations

Let $\mathbb{R}$ be a field of real numbers and $\mathbb{R}^n$ be a space with the real vectors of dimension $n$ with the Euclidean norm $||\alpha||$. Let $\mathbb{Z}$ be the ring of integers and $\mathbb{Z}^n$ the set of vectors in the space $\mathbb{R}^n$ with integer coordinates. The lattices considered here contain vectors with integer coordinates and the lattices are defined by modulo $q$ where $q$ is a large integer.

In Ajtai's work [12], three hard lattice problems are considered for lattices in $\mathbb{Z}^n$. It is shown that there is a probabilistic algorithm that solves them with a high probability (around 1) in polynomial time under certain conditions. These problems are,

1.  For an $n$-dimensional lattice $L$, approximately find the length of the shortest non-zero vector up to a polynomial factor;
2.  For an $n$-dimensional lattice $L$ with $v$ as a unique shortest vector, where the other vectors of length at most $n^c||v||$ are parallel to $v$ where constant $c$ has a large absolute value, find this shortest non-zero vector;
3.  For an $n$-dimensional lattice $L$, find a basis $B = \{b_1, \ldots, b_n\}$ with the smallest length up to a polynomial factor where the length is $\max_{i=1}^{n} ||b_i||$ .

In the work of Ajtai [12], they presented a class of problems for a certain random lattice class with the characteristics we mentioned above to find a short vector. The solution to these problems implies the solution for the above-mentioned three hard lattice problems. The thinking behind this is that if there is an algorithm that solves a random instance of a hard problem with high probability, then there is an algorithm that solves the hard problem in the worst-case scenario.

As a result, reductions for the worst-case scenarios of the lattice problems were presented. Additionally, as a corollary, if there is no probabilistic solution that solves any of the above hard lattice problems in polynomial time, then there exists a one-way lattice function for these problems. This was introduced as the "short integer solution" with the one-way function and was proven to be hard to solve. In addition, a cryptographic function for lattices based on worst-case complexity assumptions was introduced in this work. This Ajtai function and the "short integer solution" problem are heavily applied in lattice-based cryptographic approaches.

2.3.3. Desirable Features of Lattice-Based Cryptography

Lattice-based cryptographic approaches are considered to be promising post-quantum cryptographic schemes because they have many attractive features and have been explored by many researchers. In this section, several of these features will be discussed.

- **Security guarantees due to the hardness of worst case scenarios of lattice problems:** Cryptographic schemes require problems that are hard to solve in the average scenario. Ajtai [12] presented proof that randomly generated average cases of popular lattice problems are hard to solve if worst cases of the related lattice problems are hard to

solve. He gave a one-way function for the "short integer solution" and proved it is as hard as solving the worst cases of the hard lattice problems. This proof enables cryptographers to use these hard lattice problems in cryptographic schemes. At present, Ajtai's findings have been improved and applied in many cryptographic schemes ranging from plain lattices [13] to lattices with special algebraic structures [14].

- **Security against quantum computer based attacks:** Since currently used public-key cryptographic schemes that are based on the hardness of integer factorization or the discrete logarithm problem such as the RSA [1] and the Diffie–Hellman key exchange protocol [2] can be efficiently solved by Shor's quantum algorithm [3], these schemes might be insecure in the near future. Currently, an efficient algorithm based on either classical or quantum-based computers which solve lattice-based problems used for cryptography is not yet known.

- **Less computationally cost operations which can be paralleled:** The mathematical operations behind the lattice-based cryptography are more simple when considering the currently used Diffie–Hellman protocol [2] and the RSA cryptosystem [1]. The lattices are represented as multi-dimensional matrices. Therefore, the underlying operations are simple matrix operations such as addition, subtraction, and multiplication. Additionally, the elements of the lattices are modulo small integers of 10–12 bits. Since these operations are vector/matrix operations, they are highly parallelizable, and threaded concurrency can be applied for implementations to improve efficiency and speed.

- **Fully homomorphic encryption:** The first fully homomorphic encryption was introduced by Craig Gentry [15], allowing us to do computations over ciphertexts while preserving the secrecy of the plain text. This construction uses hard lattice problems of lattices. This can be mainly applied to search engines, machine learning, and medical applications (or other sensitive applications) and enables us to implement privacy-preserving systems. The reasons for using lattices for this are the low complexity of the decryption algorithm in lattice-based encryption compared to schemes such as the RSA [1] and the security of the scheme can be based on hard problems on the lattices. Additionally, ideal lattices inherit addition and multiplication operations from polynomial rings. This also makes lattice-based cryptography more suitable for fully homomorphic encryption.

- **Reasonable key sizes:** In lattice-based cryptography, multi-dimensional vectors are used as keys. These vectors consist of integers reduced by the modulo operation. Therefore, the sizes of the integers are small. Although the sizes are not as small as those currently used for classical cryptosystems, the key sizes are small enough for practical applications. Public key exchange can also be improved using a pseudo-random number generator (PRNG) [16]. Instead of sharing the public key, PRNG and a small key are used to generate the public key. This improves the efficient use of resources in the process.

- **Diverse applications:** Apart from use as a public-key cryptographic scheme, lattice functions can be used as one-way trapdoor functions [17] which enables efficient recovery of the plain text given the trapdoor information. Additionally, these lattice-based trapdoor functions enable digital signature and identity-based encryption schemes where a user generates a public key from a unique identifier (identity) and the private key is generated by the trusted third party called a private key generator (PKG) using the public key. Therefore, public key distribution prior to the exchange of ciphertext is not required. This implementation also relies on the hardness of the lattice-based problems.

## 3. Learning with Errors (LWE)

Since Ajtai [12] introduced the "short integer solution" (SIS) on lattices, lattice-based cryptography research has seen sharp progress. For example, public-key encryption [13], identity based encryption [18], and fully homomorphic encryption [15] can be identified.

With the SIS, the learning with errors (LWE) problem on lattices is applied in these cryptographic schemes. The LWE is provably hard as the popular lattice problems in the worst case, and this makes it a prime candidate for lattice-based cryptographic approaches along with the SIS.

Currently, there are no known quantum-based or classical algorithms that will solve the LWE problem in polynomial time. Therefore, it is a promising candidate for lattice-based post-quantum cryptographic schemes. Another important factor is that this can be easily implemented for classical computers. Since the LWE is a lattice-based approach, it can be implemented efficiently in instruction set architectures used by the existing computers including small embedded devices. If threaded concurrency is available then we can parallelize the operations as well.

In simple terms, the LWE problem is a set of equations represented in matrix form with some noise added to it. This makes it a system of linear equations which is harder to solve due to the added noise. This can be applied to plain lattices which only consist of integers, as well as lattices with special algebraic structures such as polynomial rings. After the initial introduction by Regev, the dual LWE cryptosystem was introduced by Gentry et al. [18]. In this section, we will focus on Regev's cryptosystem and the dual LWE cryptosystem.

*Versions of the LWE problem*

Now let us take a look at different versions of the LWE problem.

Notations

- $q$ : a large positive integer
- $s$ : a secret vector consisting of integers modulo $q (s \in \mathbb{Z}_q^n)$
- $A$ : an $n \times m$ matrix which consists of integers in modulo $q$, where $n$ is the LWE dimension and $m$ is the number of available samples. $(A \in \mathbb{Z}_q^{n \times m})$
- $e_i$ : noise drawn from small discrete Gaussian distribution centered at 0 as in Figure 2 with a standard deviation of $\sqrt{n} \ll q$ and width of $\alpha q$ where $\alpha < 1$
- $b_i$ : $b^t = s^t A + e^t \mod q (b \in \mathbb{Z}_q^n)$

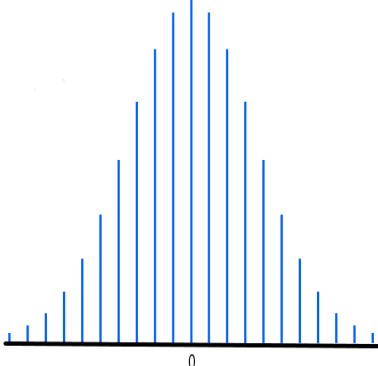

**Figure 2.** Discrete Gaussian distribution centered at 0.

When considering the LWE problem, there are two versions considered for cryptographic implementations, i.e., (1)—search and (2)—decision.

Search

Find the secret vector $s \in \mathbb{Z}_q^n$ when the attacker has access to many independent samples of noisy inner products of $s$. The number of samples available in $m$ should be sufficient to uniquely define the secret $s$ with high probability. Consider $A = \{a_1, a_2, \ldots, a_m\}$ where, $a_i \in \mathbb{Z}_q^n$ vectors are the columns of the matrix. Then generate,

$$b_1 = (\langle s, a_1 \rangle + e_1) \mod q$$
$$b_2 = (\langle s, a_2 \rangle + e_2) \mod q$$
$$\vdots$$
$$b_m = (\langle s, a_m \rangle + e_m) \mod q \tag{1}$$

Equation (1) represents the generation of noisy inner products of $s$. The problem is that the attacker should find $s$, given $\{b_1, b_2, \ldots, b_m\}$ and $A$.

$$b^t = s^t A + e^t$$

If there is no noise or if the noise is known, an attacker can easily solve this problem because the inner product of $s$ and $a_i$ would be exactly $b_i$, and then by using basic algebraic operations, it can be solved by Gaussian elimination.

Decision

Given the $m$ independent samples of $(a_i, b_i)$ pairs, the challenge is to distinguish between the $(a_i, b_i)$ pair that was generated as a noise-added inner product and a uniformly random pair $(a_i, b_i)$. In the case of a uniformly random pair, $b_i$ is completely independent of $a_i$ and there is no noise $e_i$ as well.

### 3.1. Regev's LWE Cryptosystem

We discuss Regev's [13] findings in this section. He proved the worst-case hardness for search-LWE and decision-LWE problem versions. Moreover, he introduced the first public-key cryptosystem based on the hardness of the LWE problem, which is often known as Regev's LWE cryptosystem. Recall the notations introduced in Section 3. Encryption/decryption of a single bit is considered.

The main parameters of this cryptosystem are the LWE dimension $n$, the number $m$ of samples required for correct decryption, and the Gaussian error distribution $\mathcal{X}$. For successful decryption, $m \approx (n+1) \log q$ is required. Decryption is accurate only if the accumulated error $E_T$ is less than $\frac{q}{4}$. This can be ensured by choosing $q$ to be significantly larger than $m$ and all the values in the error distribution $\mathcal{X}$.

#### 3.1.1. Key Generation

- **Private Keys**

    - $s^t \leftarrow \mathbb{Z}_q^n$ ($n$ length vector with modular $q$ integer entries. This is mainly considered as a secret known only to the receiver, rather than a key)

- **Public Keys**

    - $A \in \mathbb{Z}_q^{n \times m}$
    - $b^t = s^t A + e^t$ ($e$ is drawn from the discrete Gaussian error distribution of $\mathcal{X}$)

Both the private and public keys can be represented as matrices. The size of the secret/private keys is $O(n)$ and the size of the public keys is $O(n^2)$, where $n$ is the number of bits. Matrix multiplication and addition operations are mainly used in this process. Moreover, matrix/vector generation operations are also used. These operations can be parallelized to increase the speed. $A$ contains $n \times m$ integers and might require many resources; therefore, sharing $A$ distributively can be useful. Using a pseudo-random number generator (PRNG) [16], a smaller seed can be generated and shared, in a way that the seed is much smaller than $A$. Then, the receiver can generate $A$ using the seed. Noise $e$ is generated from a Gaussian distribution.

### 3.1.2. Encryption

Encryption is carried out using the public key $b^t$, and the ciphertext preamble is generated by $A$.

Steps

1. The sender obtains the public key $(A, b^t)$;
2. The sender calculates $m$-length binary vector $x \leftarrow \{0, 1\}^m$;
3. The sender calculates the ciphertext preamble $u = Ax$;
4. The sender encrypts a `bit` and obtains the ciphertext scalar $u' = b^t x + \texttt{bit} \cdot \lfloor \frac{q}{2} \rfloor$ and sends the payload $(u, u')$ to the receiver

These operations can be highly parallelized. The ciphertexts are in $\Omega(n)$.

### 3.1.3. Decryption

Decryption is done by using the private key $s^t$.

Steps

1. The receiver obtains the payload $(u, u')$;
2. The receiver calculates the scalar $r = u' - s^t u$;
3. The receiver evaluates whether $r \approx \frac{q}{2}$ is true or false;
4. If true, then the plain text bit is 1, else 0.

These computations can be parallelized as the previous key generation and encryption operations. All the results of these steps are in modulo $q$. To encrypt and decrypt multiple $k$-bits, we need to modify the dimensions of the payload $(u, u')$ and the other vectors associated with encryption and decryption to hold data for $k$-bits. The modifications are listed in Table 1.

**Table 1.** Dimension modification required for Regev's multi-bit encryption/decryption scheme.

| Scalar/Vector | Dimensions for 1-bit Encryption | Dimensions for $k$-bit Encryption |
|:---:|:---:|:---:|
| $u$ | $n \times 1$ | $n \times k$ |
| $u'$ | $1 \times 1$ | $1 \times k$ |
| $r$ | $1 \times 1$ | $1 \times k$ |
| $x$ | $m \times 1$ | $m \times k$ |

### 3.2. Dual LWE Cryptosystem

Now we discuss the dual LWE cryptosystem [18] of Gentry et al., which is based on Regev's LWE problem. Recall the notations introduced in Section 3. Encryption/decryption of a single bit is considered.

The main parameters of this system are the same as Regev's LWE cryptosystem. The number of samples $m$ should be selected according to $m \approx n \log q$ for accurate decryption. In Regev's cryptosystem, nonuniform distribution is used to generate public keys and have a unique secret key. However, for a given public key, there exist many encryption randomness choices to generate the same ciphertext. Differently, public keys of the dual cryptosystem are generated from statistically random distributions and a public key has many possible secret keys. The encryption randomness which generates a ciphertext is considered unique. This is very useful for the implementation of advanced cryptosystems.

### 3.2.1. Key Generation

Here we explain about the private and public keys of the system.

- **Private Keys**

  – $x \leftarrow \{0, 1\}^m$ ($m$ length binary vector)

- **Public Keys**
  - $A \in \mathbb{Z}_q^{n \times m}$
  - $u = Ax$

Both private and public keys can be represented in matrix form. The size of the secret/private keys is $O(n)$ and the size of the public keys is $O(n^2)$ where $n$ is the number of bits in the key. Matrix multiplication and addition operations are mainly used in this process in addition to matrix/vector generation operations. These operations can be parallelized to increase the speed of the operations. $A$ can be shared efficiently using a PRNG [16] as in Regev's cryptosystem. Noise $e$ is generated from a Gaussian distribution as before for security requirements.

### 3.2.2. Encryption

Encryption is performed with the public key $u$ and the ciphertext preamble is generated using $A$.

Steps

1. The sender obtains a public key $(A, u)$;
2. The sender generates vector $s \leftarrow \mathbb{Z}_q^n$;
3. The sender calculates the ciphertext preamble $b^t = s^t A + e^t$ where $e$ is drawn from the discrete Gaussian error distribution $\mathcal{X}$;
4. The sender encrypts a `bit` and get the ciphertext scalar $b' = s^t u + e' + \texttt{bit} \cdot \lfloor \frac{q}{2} \rfloor$ and sends the payload $(b^t, b')$ to the receiver ($e'$ is drawn from $\mathcal{X}$ discrete Gaussian error distribution).

These operations use matrix multiplication and addition which can be parallelized. The ciphertexts have size $\Omega(n)$.

### 3.2.3. Decryption

The decryption is done by the private key $x$.

Steps

1. The receiver obtains the payload $(b^t, b')$;
2. The receiver calculates the scalar $r = b' - b^t x$;
3. The receiver evaluates whether $r \approx \frac{q}{2}$ is true or false;
4. If true, then the plain text bit is 1, else 0.

As in Regev's cryptosystem, matrix operations in the dual cryptosystem can be parallelized. All computations of these steps are done in modulo $q$. To encrypt and decrypt multiple $k$-bits, we need to modify the dimensions of the payload $(u, u')$ and the other vectors associated with encryption and decryption to hold data for $k$-bits. The modifications are listed in Table 2.

**Table 2.** Dimension modification required for Regev's multi-bit encryption/decryption scheme.

| Scalar/Vector | Dimensions for 1-bit Encryption | Dimensions for $k$-bit Encryption |
|:---:|:---:|:---:|
| $b^t$ | $n \times 1$ | $n \times k$ |
| $b'$ | $1 \times 1$ | $1 \times k$ |
| $r$ | $1 \times 1$ | $1 \times k$ |
| $x$ | $m \times 1$ | $m \times k$ |

## 4. Ring-LWE

Lyubashevsky et al. [14] introduced ring-LWE as another LWE-based scheme that can be applied to implementing public-key cryptographic schemes. In this work, they have given estimations for the hardness of ring-LWE-based problems as well. The main reason

behind this was to answer the open question of whether the LWE can be made more efficient by introducing extra algebraic structures. This question arose because of the shortcomings in terms of efficiency and memory usage in plain LWE-based implementations. To encrypt or decrypt a single bit in plain LWE, there need to be more than $n^2$ operations. Since $n$ needs to be sized either 100 or 1000 for hardness, this is highly inefficient. Moreover, the cipher texts are in $\Omega(n)$.

In order to increase the efficiency, instead of considering the matrix multiplication, which takes $O(n^2)$ operations, multiplication in the polynomial ring is introduced. This takes around $n \log n$ operations using fast Fourier transformations. Therefore, unlike the inner product $b_i = (\langle s, a_i \rangle + e_i) \mod q$, which generates scalar $b_i$, operation in the ring-LWE is done as multiplication of two polynomials that results in a polynomial. Further, the discrete Gaussian distribution that is added here becomes an $n$-dimensional vector. These operations can be completed in linear time.

### 4.1. Ring Structure

Let $R = \frac{\mathbb{Z}[X]}{\langle X^n + 1 \rangle}$ be an integer polynomial ring in modulo $X^n + 1$ where $X^n + 1 \in Z[X]$ and the security parameter $n$ is a power of 2, thus making $X^n + 1$ irreducible over rationals. Therefore, elements in the ring $R$ are the set of integer polynomials with a degree less than $n$.

### 4.2. Versions of Ring-LWE

As in plain LWE, there are two similar versions of ring-LWE as well, i.e., (1)—search and (2)—decision.

#### 4.2.1. Search

Find the secret ring element $s \in R_q$ when the attacker has access to many independent samples of noisy random ring products of $s$. Unlike in the plain LWE, here the noise is also a polynomial that needs to be added to each coefficient of the ring-product polynomial. As in the plain LWE version, the number of available samples $m$ should be sufficient to uniquely define the secret $s$ with high probability.

$$
\begin{aligned}
a_1 &\leftarrow R_q, \ b_1 = (\langle s, a_1 \rangle + e_1) \mod q \in R_q \\
a_2 &\leftarrow R_q, \ b_2 = (\langle s, a_2 \rangle + e_2) \mod q \in R_q \\
&\vdots \\
a_m &\leftarrow R_q, \ b_m = (\langle s, a_m \rangle + e_m) \mod q \in R_q
\end{aligned}
\tag{2}
$$

Equation (2) represents the generation of random ring products of $s$. Given $\{b_1, b_2, \ldots, b_m\}$ and $A$, attacker should find $s$. If there is no error polynomial and $A = \{a_1, a_2, \ldots, a_m\}$ is invertible, this can be solved easily. Often $A$ is invertible.

#### 4.2.2. Decision

Given the $m$ number of independent samples of $(a_i, b_i)$ pairs, the challenge is to distinguish between the $(a_i, b_i)$ pair that was generated as a noise-added random ring-product and a uniformly random $(a, b_i) \in R_q \times R_q$. In the case of a uniformly random pair, $b_i$ is completely independent of $a_i$ and there is no noise $e_i$.

### 4.3. Ring-LWE-Based Public-Key Encryption Scheme

Recall Regev's cryptographic scheme in Section 3.1. In this section, the ring-LWE version of it is considered. This system is well analyzed in Lyubashevsky et al. [19]. They have given an application for a public-key cryptosystem where each ciphertext and public key consist of two ring elements. This is an extension of the public-key-based cryptosystem on plain lattices. The ring is fixed as $R = \frac{Z[X]}{(X^n + 1)}$.

### 4.3.1. Key Generation

Here we explain about the private and public keys of the system.

- **Private key**
    - $s \in R_q$ ($n$ length secret vector with modulo $q$ integer entries in ring $R_q$ where $q \in \mathbb{Z}^+$)
- **Public key**
    - $(a, b = a \cdot s + e) \in R_q^2$ (where $a \in R_q$ and random small elements $s, e \in R$ from the error distribution)

### 4.3.2. Encryption

Consider the encryption of $n$-bit message $z$ .

Steps

1.  $n$-bit message is seen as an element of $R$ and the bits are used as coefficients of a polynomial of degree less than $n$;
2.  generate $e_1, e_2, r \in R$ from error distribution;
3.  Calculate $u = a \cdot r + e_1$;
4.  Calculate $v = b \cdot r + e_2 + z \cdot \frac{q}{2}$ and send the payload $(u, v) \in R_q^2$ to receiver.

### 4.3.3. Decryption

Steps

1.  Receive the payload $(u, v) \in R_q^2$;
2.  Calculate $r = v - u \cdot s$;
3.  Evaluate each $r_i \approx \frac{q}{2}$;
4.  If $r_i \approx \frac{q}{2}$, then $r_i = 1$, else 0.

The process is almost similar to Regev's scheme. The difference is replacing the inner products with the ring products, hence resulting in new ring structures. This increases the efficiency of the operations.

## 5. Lattice Trapdoors

### 5.1. Basic Definitions and Operations

A function that is easy to evaluate but hard to invert on its own is called a trapdoor function. However, the inversion of this function can be made easy by extra trapdoor information generated in the beginning. There are many versions of trapdoor functions. For example, in the RSA algorithm [1], we can identify the factorization of $n$ to $p$ and $q$ as a trapdoor, because by using $p$ and $q$, $d$ can be calculated. Similarly, we can identify lattice trapdoors. A simple representation of a lattice trapdoor is shown in Figure 3, where $f$ is the trapdoor function and $t$ is the trapdoor information required to invert and get input $x$. There are two distinct notions used for lattice trapdoors, i.e., (1)—short-basis trapdoors, and (2)—gadget-based trapdoors.

1.  **Short basis trapdoors:** This lattice is made up with the basis of relatively short lattice vectors. The short-basis lattices can be served as trapdoors for lattices with bad basis.
2.  **Gadget-based trapdoors:** This applies only for $q$-ary lattices that arise from (ring)-SIS/LWE problems. These specially structured matrices $G$ are called "gadgets". They solve the underlying SIS and LWE problems easily.

When considering the aforementioned two notions, comparatively, gadget-based trapdoors are simpler to work with in SIS/LWE-based applications. Moreover, they are more efficient. Therefore, most of the lattice-based cryptographic schemes use gadget-based trapdoors. There are various applications of lattice trapdoors; they are used to build identity-based encryption schemes, eliminate random oracles, construct signature schemes, etc. Following, we focus on gadget-based trapdoors.

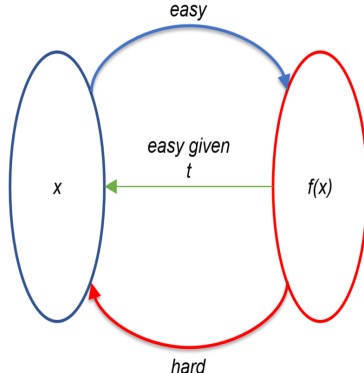

**Figure 3.** Representation of a lattice trapdoor.

### 5.1.1. Gadget-Based Trapdoors

Here, a vector called $g$ is used to build a gadget matrix $G$. The $g$ can be defined as below, where $q = 2^k$.

$$g = \{1, 2, 4, \ldots, 2^{k-1}\} \in \mathbb{Z}_q^{1 \times k}$$

By using $g$, the gadget matrix $G$ can be obtained as below,

$$G = I_n \otimes g = \begin{bmatrix} \cdots g \cdots & & & \\ & \cdots g \cdots & & \\ & & \ddots & \\ & & & \cdots g \cdots \end{bmatrix} \in \mathbb{Z}_q^{n \times nk}$$

According to the LWE problem, the LWE function can be described as below.

$$g_A(s, e) = sA + e$$

To find $s$ of the above equation, $g_A^{-1}$ should be evaluated. Since it is hard to directly evaluate $g_A^{-1}$, this can be identified as a trapdoor. To make it easier to obtain $g_A^{-1}$, $A$ can be represented in a different form with the help of a gadget matrix. For that, a semi-random matrix can be defined as $[\bar{A}|G]$ for uniform $\bar{A} \in \mathbb{Z}_q^{n \times \bar{m}}$. By choosing a short (Gaussian) $R \leftarrow \mathbb{Z}^{\bar{m} \times n \log q}$, it can be used as the required trapdoor information to evaluate $g_A^{-1}$. The matrix $A$ can be defined as below by using the gadget matrix $G$.

$$A = [\bar{A}|G] \begin{bmatrix} I & -R \\ & I \end{bmatrix} = [\bar{A}|G - \bar{A}R]$$

By representing $A$ as above, $g_A^{-1}$ can be used as $g_G^{-1}$. By implementing inversion using the above representation, a gadget-based trapdoor application can be constructed. It can be reduced $g_G^{-1}$ to $n$ parallel calls to $g_g^{-1}$. Let,

$$S = \{s_1, \ldots, s_n\}, \qquad e = \{e_1, \ldots, e_n\}$$

Then, $g_G$ can be written as below,

$$g_G(S, e) = SG + e = (s_1 g + e_1, \ldots, s_n g + e_n)$$

Each term in $g_G$ can be considered as follows,

$$g_g(s, e) = sg + e$$

Therefore, by finding the $g_g^{-1}$ of each term, $g_G^{-1}$ can be obtained. This operation can be parallelized, the because inverse of each term can be obtained separately. Let,

$$b = \{b_0, b_1, \ldots, b_{k-1}\} = sg + e = (s + e_0, 2s + e_1, \ldots, 2^{k-1}s + e_{k-1}) \mod q$$

By recovering the binary digits $s_o, s_1, \ldots, s_{k-1}$ of $s \in \mathbb{Z}_q$, beginning from the least significant bit to the most significant bit, $s$ can be recovered. First, the $s_0$ (LSB) of $s$ can be determined as below.

$$b_{k-1} = 2^{k-1}s + e_{k-1} = (q/2)s_0 + e_{k-1} \mod q$$

By testing whether $b_{k-1}$ is closer to 0 or $q/2(\mod q)$, $s_0$ can be determined. Then, $s_1$ can be determined as below.

$$b_{k-2} = 2^{k-2}s + e_{k-2} = 2^{k-1}s_1 + 2^{k-2}s_0 + e_{k-2} \mod q$$

Since we know $s_0$, by subtracting $2^{k-2}s_0$ from above $b_{k-2}$ and testing the result value for 0 or $q/2$, $s_1$ can be determined. Likewise, all the bits of scalar $s$ can be recovered, hence $s$ can be determined. According to this by using $g_g^{-1}$, $g_G^{-1}$ can be obtained.

### 5.2. Applications of Lattice Trapdoors

In this section, we discuss some applications of lattice trapdoors: (1)—digital signature schemes, and (2)—public-key encryption schemes.

### 5.2.1. Digital Signature Schemes

A digital signature scheme is used for the verification of the authenticity of a digital message. In Internet communication, a method of establishing the authenticity of messages is a major requirement for information security. Digital signatures are integrated into public-key cryptographic schemes themselves. In some countries, such as EU member states, digital signatures are legally valid.

In public-key-based digital signature schemes, a trusted third party signs the certificate that carries the public key. Therefore, anyone can trust the ownership of the public key. The sender can sign a message using the private key and the receiver can verify the ownership of the message using the sender's public key. Most of the existing public-key-based digital signature schemes are based on either the RSA scheme, the Elgamal scheme, or the DSA scheme.

Let $\text{Sig} = (\text{KeyGen}, \text{Sign}, \text{Verify})$ be a digital signature scheme where KeyGen is the key generation algorithm, Sign is the signing algorithm and Verify is the signature verification algorithm. Let $M$ be an arbitrary message.

- **Key generation:** $\text{KeyGen}(1^n)$ : outputs signing key $K_s$ and verification key $K_v$
- **Message signing:** $\text{Sign}(K_s, M)$ : Sender signs message $M$ using $K_s$ and generates a digital signature $\sigma \in \{0, 1\}^*$
- **Verification:** $\text{Verify}(K_v, M, \sigma)$ : Receiver verifies the message $M$ using $K_v$ and $\sigma$. Then either accept the message or reject it depending on the success of the verification.

The work of Gentry et al. [18] introduced the application of lattice trapdoors for implementing digital signature schemes. Their objective was to implement a simple and efficient digital signature scheme based on lattices. This implementation is in the random oracle model. Micciancio and Peikert [17] introduced a statically secure digital signature scheme in the standard model. In the scheme of Micciancio and Peikert, a gadget matrix $G \in \mathbb{Z}_q^{n \times nk}$ is used for the trapdoor implementation. Their signature scheme is as follows.

Key Generation

1. Generate $\bar{A} \leftarrow \mathbb{Z}_q^{n \times \bar{m}}$ where $\bar{m} = O(nk)$ and $k = \log q$
2. Select trapdoor $R$ for $A$ such that $R \in \mathbb{Z}^{\bar{m} \times nk}$ from the Gaussian distribution $D_{\mathbb{Z}, \omega(\sqrt{\log n})}^{\bar{m} \times nk}$

3. Calculate $A = [\bar{A}|G - \bar{A}R]$
4. For $i = 0, 1, \ldots \ell$, $A_i \leftarrow \mathbb{Z}_q^{n \times nk}$ where $\ell$ is the message length
5. Select $u \leftarrow \mathbb{Z}_q^n$ where $u$ is a syndrome
6. $K_v = (A, A_0 \ldots, A_\ell, u)$ and $K_s = R$

Message Signing

1. generate $A_M = \left[A|A_0 + \Sigma_{i \in [l]} M_i A_i\right] \in \mathbb{Z}_q^{n \times m}$
2. Output: $||v|| \in \mathbb{Z}^m$ from the distribution generated with $A_M$ using trapdoor $R$ for $A$

Signature Verification

1. Check whether $v \leq s \cdot \sqrt{m}$ and $A_M \cdot v = u$
2. Accept if both the conditions are true, else abort

As in LWE-based encryption schemes, the matrix operations in this signature scheme can be parallelized to increase efficiency.

### 5.2.2. Chosen-Ciphertext Attack (CCA) Secure Public-Key Encryption Schemes

CCA-secure encryption schemes are not vulnerable to chosen ciphertext attacks (CCA). In this category of attacks, the adversary has access to the plaintexts that correspond to its choice of ciphertexts. The CCA-secure encryption schemes can be built using gadget-based trapdoors. Here, to compute the secret key, the gadget matrix is used; this scheme is built using $R$ (refer to Section 5.1.1) as the secret key.

Let PKE = (KeyGen, Enc, Dec) be a public-key encryption scheme where KeyGen is the key generation algorithm, Enc is the encryption algorithm and Dec is the decryption algorithm. Let $m$ be an arbitrary message.

Basic Operations

- **Key generation:** KeyGen($1^n$): outputs the public key $A$ where $A = [\bar{A}|A_1]$ and $A_1 = -\bar{A}R \mod q$ and secret key as $R$
- **Encryption:** Enc($pk, m$) : outputs the ciphertext using the encryption algorithm and the public key for message $m$
- **Decryption:** Dec($sk, c$): outputs the resultant plain text using the decryption algorithm with the secret key $R$ and the ciphertext

## 6. LWE-Based Cryptographic Implementations

There are many implementations of lattice-based cryptosystems using the LWE problem and its variant learning with rounding (LWR). The National Institute of Standards and Technology (NIST) is selecting public-key cryptographic algorithms that are secure against quantum attacks. The search for post-quantum cryptosystems began in November 2017, and a total of 82 candidates submitted their proposals. Currently, the NIST has finished the third round and selected a key encapsulation mechanism (KEM) for standardization, which is known as "CRYSTALS-Kyber". Moreover, there are another four candidate algorithms selected for the fourth round.

In the following we discuss some schemes that were selected through the second round of the NIST post-quantum cryptography standardization process.

### 6.1. CRYSTALS-Kyber

Kyber [20] is a part of CRYSTALS—Cryptographic Suite for Algebraic Lattices, which is a package submitted to NIST in 2017. This is based on the module LWE problem. Kyber offers a CCA-secure KEM which started as a CPA-secure public-key encryption scheme and then transformed into a CCA-secure KEM, by applying a variant of the Fujisaki–Okamato transform. Based on the CCA-secure KEM, Kyber provides CCA-secure encryption and authenticated key exchange. The authors claim that Kyber has an efficiency similar to the ring-LWE-based schemes with increased security and flexibility. Kyber can change the

level of security by changing a single parameter. Kyber has two sets of parameters for long-term security and short-term security. Regarding security, the authors have found that the dimension of the modulus is a non-trivial factor when the scheme is under attack. They have stated that the other ring-LWE and NTRU-based schemes are more susceptible to attacks than module-LWE-based schemes. Kyber is currently chosen by *NIST* for standardization.

### 6.2. FrodoKEM

FrodoKEM [21] is a KEM based on unstructured (plain) lattices. It uses the plain LWE problem to ensure security. Since the lattice is unstructured, the key sizes of FrodoKEM are larger compared to the structured lattices. Although the key sizes are large, the security of FrodoKEM is higher compared to the structured lattice-based schemes because it does not have any extra information that an attacker could exploit. Moreover, FrodoKEM uses pseudorandom number generators to generate large matrices and the only need is to share the seed. Thus, it reduces the bandwidth of key sharing significantly.

### 6.3. LAC

LAC [22] is a public-key encryption scheme based on the ring-LWE problem. The lattices with the ring structures carry out their multiplication operations by either using number theoretic transformation (NTT) or fast Fourier transformation (FFT). However, to use the NTT algorithms, there is a minimum module limitation. The $q$ should be equal to or higher than 12289. Since this limitation is only dictated by the choice of NTT, the LAC authors have considered using a smaller modulus. They used a byte-size modulus and a constant ratio between the error and the modulus. They faced challenges such as decryption errors and efficiency issues, whereas they have found solutions with parameter customization and heavy-error correction codes. The most notable thing about LAC is its compactness due to the byte-size modulus.

### 6.4. NewHope

NewHope [23] is a KEM based on the ring-LWE problem. Their work is inspired by another implementation of Bos et al. [24]. The authors of NewHope have analyzed the scheme and proposed new parameters and better error distribution. According to the authors, the very large modulus and the Gaussian sampler are reasons for the inefficiency of the scheme. They have stated that "High quality Gaussian noise is crucial for encryption based on LWE" is a misconception and it has made other implementations slower.

NewHope has reduced the modulus from $2^{32}$ to $2^{14}$, increasing both security and efficiency. Moreover, they have done a detailed security analysis against quantum attacks. Regarding the noise distribution, they have replaced the discrete Gaussian distribution with a centered binomial distribution. In order to increase the security of the scheme, they do not rely on globally chosen public parameters and they do not perform short-term key caching.

### 6.5. Round5

Round5 [25] is a public-key encryption scheme based on the ring-LWE rounding (RLWR). It is a combination of Round2 and HILA5. From Round2, it inherits the RLWR as the base problem which results in a larger design space. From HILA5, it uses XEf for error correction. This results in a low failure rate, shrunk parameters, and increased security and performance. According to the NIST, Round5 is among the lattice-based schemes with the lowest bandwidth requirements. With the new parameters, it has failure rates below $2^{-128}$.

### 6.6. Three Bears

Three Bears [26] is a key exchange and encryption scheme based on mLWE (module-LWE). Instead of choosing a usual polynomial ring, Three Bears uses generalized Mersenne rings. They have focused on minimizing bandwidth costs and maximizing security. The authors have chosen three sets of parameters (BABYBEAR, MAMABEAR, PAPABEAR).

They have studied Saurinen's error correction trick for unauthenticated key exchange. In the implementation of the scheme, it has used a low signal-to-noise ratio, to achieve minimum bandwidth and maximum security. However, it is done with the risk of key exchange failure. It has a $2^{-55}$ key exchange failure probability. These failures leave the scheme vulnerable to chosen-ciphertext attacks. To reduce this key failure probability, the authors have shown interest in error correction codes instead of increasing the signal-to-noise ratio. According to the authors, it is not possible to reduce the failure probability to the cube of the current probability. The authors have concluded that the Mersenne rings are capable of replacing polynomial rings for lattice problems.

*6.7. FALCON*

Falcon [27] is a lattice-based signature scheme. It is an implementation of the theoretical framework of Gentry et al. [18]. They have chosen the NTRU lattices and fast Fourier sampling. Their primary goal is to minimize the public key and signature sizes. The NTRU lattices speed up computations and the ring structure reduces the key size. This implementation introduces a recursive algorithm for Gaussian sampling using the "Falcon Tree" data structure.

In addition to the compact nature of the Falcon, it provides fast signature generation and verification capabilities. However, it has some limitations as well; according to the authors, the implementation of fast Fourier sampling and key generation is relatively hard to understand and complicated to implement. They have justified this by saying fast Fourier transform and trees are familiar to most developers.

*6.8. qTESLA*

qTesla [28] is a family of post-quantum digital signature schemes. It focuses on simplicity and security. It is a portable scheme with constant time operations. Moreover, it has provided assembly implementations. qTesla is based on the ring-LWE problem. The new features in qTesla can be summarized into simplicity and security. The qTesla authors take pride in their simple and compact implementation that supports more than one security level and more than one parameter set. However, the NIST has some comments about the security and performance of the qTesla implementation. The NIST has found an issue with the parameter set which affects the performance of the scheme.

## 7. Comparisons between Lattice-Based Cryptosystems and Implementations

In this section, we summarize and compare the characteristics of different LWE-based cryptosystems as well as the currently available LWE-based cryptographic implementations.

*7.1. Comparison between Major Plain LWE Cryptosystems*

Currently, the major plain LWE-based approaches are Regev's cryptosystem [13] and the dual cryptosystem [18]. Even though these two systems mostly have the same types of internal operations, there exist a couple of significant differences that are unique to each other. They are:

- When considering the keys in Regev's system, the public key $(A, b^t)$ is pseudo-random and the secret key $s$ is unique to a particular public key. Whereas, in the dual system, the public key $(A, u)$ is statistically random and there is a possibility to have multiple secret keys $x$.
- In Regev's system, to produce a ciphertext, there are many ways to introduce encryption randomness. Whereas, in the dual system, the encryption randomness is unique.

Table 3 summarizes the comparison between the major plain LWE cryptosystems.

**Table 3.** Comparison between major plain LWE cryptosystems.

| Cryptosystem | Public Key | Secret Key | Encryption Randomness | Parallelizability |
|---|---|---|---|---|
| Regev [13] | pseudo-random | unique to a public key | many ways to achieve | encryption, decryption |
| Dual [18] | statistically random | can have multiple | unique to a ciphertext | encryption, decryption |

### 7.2. Comparison between Ring-LWE and Plain LWE Cryptosystems

The ring-LWE problem has several attractive features over the plain LWE problem for cryptographic implementations. By using fast Fourier transformations (according to the paper of Lyubashevsky et al. [19]), polynomial multiplication can be done in $O(n \log n)$ operations which is much more efficient than the inner-product operations. This reduces encryption/decryption time significantly.

In the ring-LWE, $(a, b) \in R_q \times R_q$ replaces $(\mathbf{a}, b) \in \mathbb{Z}_q^n \times \mathbb{Z}_q$ in standard-LWE methods. ($\mathbf{a}$ in standard LWE is a vector) This shortens the public key size by a factor of $n$. This often reduces the secret key size by the same factor as well. Reducing the public key size is highly beneficial for key exchange purposes. The large size of the public key has been a serious drawback of the plain LWE cryptographic schemes. This too results in the increased efficiency of underlying operations in the cryptosystem. The security of the system of ring-LWE is based on the worst-case hardness assumptions on ideal lattices. As mentioned before, there is no known quantum or classical computer-based algorithm to solve these problems with polynomial time complexity.

However, since ring-LWE contains a special algebraic structure, it can be used by attackers for breaking the cryptosystem. In the work of Elias et al. [29], they show that the samples of the polynomial ring $R = \frac{Z[X]}{P(n)}$ ($P(n)$ is a polynomial) are vulnerable to distinguishing attacks if a root in the $P(n)$ is small order modulo $q$. This implies that an adversary can utilize the extra polynomial ring structure in the ring-LWE to find vulnerabilities in the system. Since the plain LWE problem does not have this special structure, this vulnerability is not in it. Overall, the plain LWE applications are known to be more secure than the ring-LWE applications due to the aforementioned reason.

Recently, there have been some attempts to introduce more secure versions of the ring-LWE problem such as twisted ring-LWE [30]. There are new algorithms to increase the computational efficiency and reduce the bandwidth of the LWE operations: discrete trigonometric transform (DTT) and generalized discrete Fourier transform (GDFT) algorithms [31], calculating inner products as a branching program [32], and "decompose-and-reduce" modular multiplication algorithm (DARM) [33] to improve modular multiplication with NTT.

Table 4 summarizes the comparison between the plain and ring-LWE cryptosystems.

**Table 4.** Comparison between plain and ring-LWE cryptosystems.

| Cryptosystem | Encryption/Decryption Operation | Operation's Computational Complexity [Key] | Versions | Vulnerability |
|---|---|---|---|---|
| Plain LWE [12,13,18] | matrix multiplication | $O(n^2)$ [1] | search, decision | - |
| Ring-LWE [14] | multiplication in a polynomial ring | $O(n \log n)$ | search, decision | special algebraic structure [2] |

[1] There are new algorithms to increase the computational complexity [31–33]. [2] Elias et al. [29] presents an attack on ring-LWE. Twisted ring-LWE [30] is a secure version of ring-LWE. [key]: $n$ is the size of the secret key.

### 7.3. Comparison between the Existing LWE-Based Cryptographic Implementations

Table 5 gives a comparison between the characteristics of the existing LWE-based cryptographic implementations.

**Table 5.** Lattice type and the underlying problem of currently implemented cryptography schemes.

| Scheme | Lattice Type | Base Application | Note |
|---|---|---|---|
| CRYSTALS-Kyber [20] | Module-LWE | KEM | efficiency similar to the ring-LWE schemes |
| FrodoKEM [21] | Plain-LWE | KEM | large key sizes<br>high security comp. to the structured lattice schemes |
| LAC [22] | Poly-LWE | Public-key encryption | compact |
| NewHope [23] | Ring-LWE | KEM | increased security and efficiency of Bos et al. [24] |
| Round5 [25] | Ring-LWR | Public-key encryption | lowest bandwidth requirement<br>low failure rate |
| Three Bears [26] | Module-LWE | Public-key encryption | low bandwidth requirement<br>not possible to reduce the failure probability |
| FALCON [27] | NTRU lattice | Signature | complicated to implement<br>compact, efficient |
| qTesla [28] | Ring-LWE | Signature | issue with the parameter set<br>simple and compact implementation |

## 8. Discussion

Lattice-based cryptography has seen rapid progress with the findings of Ajtai [12]. Currently, the major plain LWE-based approaches are Regev's cryptosystem [13] and the dual cryptosystem [18]. Their hardness depends on the hardness of the lattice problems discussed in Section 3.

Apart from the hardness of the LWE problem, the security of the LWE-based cryptographic schemes is also depending on the Gaussian distribution, which is used to sample the error distribution. Without the error distribution, the equation

$$b_1 = (\langle s, a_1 \rangle + e_1) \mod q$$
$$b_2 = (\langle s, a_2 \rangle + e_2) \mod q$$
$$\vdots$$
$$b_m = (\langle s, a_m \rangle + e_m) \mod q$$

can be easily solved by using Gaussian elimination techniques. An adversary with the ability to predict the error distribution can also solve this equation and get the secret key. Therefore, if this system is to be implemented, the error sampler must be a secure one.

The plain LWE operations are mainly matrix-based operations. The most costly operation is matrix multiplication, which is used to calculate the inner product. Moreover, public key generation has $n \times m$ integer generation operations, where $m$ is the number of available LWE samples. For the implementation of these operations, using a concurrency-based approach is the most suitable for increasing the efficiency and the speed of the matrix operations. In particular, for matrix operations, threaded concurrency would be highly applicable. Further, the Strassen algorithm [34] could be employed to implement the matrix multiplication operation.

The ring-LWE cryptographic schemes [14] which utilize extra algebraic ring structures are introduced to make LWE approaches more efficient and lightweight. The ring-LWE mainly reduces the key sizes. Key exchange and encryption processes benefit from this. Moreover, the inner-product operation is replaced by polynomial ring multiplication. These modifications have a significant increase in the efficiency of the cryptosystem. Therefore, the ring-LWE is applicable for applications with less computational power and memory. The hardness of the ring-LWE problem is not simply understood as standard LWE.

*Future Directions*

A major drawback that hinders the LWE being used in practical and lightweight applications is the key sizes. When compared with classical cryptosystems such as the RSA, which uses smaller public keys, the LWE systems have larger public keys. Mainly, the public key $A$ is a large matrix that stores integers of modulo $q$. Therefore, it takes a considerable amount of time and resources to generate it. As a future direction, it is worthwhile to try methods that can reduce the key size without reducing the security of the cryptographic system. There are LWE hardness estimator tools that are widely used and recognized by cryptographers, such as the estimator used in Albrecht et al. [35].

The security of the ring-LWE should be studied further to find out any existing vulnerabilities in the polynomial structure, and to determine whether it still be a good lightweight post-quantum cryptographic scheme. Since the plain LWE does not have special structures, an adversary cannot take the advantage of it. Therefore, the standard-LWE approach is the best available LWE scheme that currently focuses mainly on security. More improvements should be made to the efficiency of the standard LWE to a level that is comparable to the ring-LWE. This will result in an efficient cryptographic scheme that not only focuses on security but also on efficiency. Since there are many implementations based on rings and other structured lattices, unstructured lattice-based cryptography schemes should be investigated further.

According to our study, the efficiency of standard LWE approaches using plain lattice-based structures should be improved. Efficient lattice trapdoor approaches can be used for this. As mentioned above, gadget-based lattice trapdoors are more efficient and simpler to work with LWE applications. Therefore, to improve the efficiency of the plain lattice-based schemes gadget-based lattice trapdoors can be used. Moreover, the gadget-based lattice trapdoors reduce the complications of using more techniques in the applications of the lattice trapdoors; as an example, CCA-secure encryption schemes can be built without using strongly unforgeable one-time signatures [36]. We believe that by continuing further studies on lattice trapdoors, it is possible to further improve the use of plain lattice-based schemes with LWE applications and to improve their efficiency to a level which is comparable to the efficiency of the ring-LWE-based schemes.

**Author Contributions:** Conceptualization, J.A., H.B., T.W. and Y.H.; methodology, H.B., T.W. and Y.H.; formal analysis, H.B., T.W. and Y.H.; investigation, H.B., T.W. and Y.H.; resources, J.A., H.B., T.W. and Y.H.; writing—original draft preparation, H.B., T.W. and Y.H.; writing—review and editing, J.A., H.B., T.W. and Y.H.; project administration, J.A.; funding acquisition, J.A. All authors have read and agreed to the published version of the manuscript.

**Funding:** The APC was funded by the Rabdan Academy.

**Institutional Review Board Statement:** Not applicable.

**Informed Consent Statement:** Not applicable.

**Data Availability Statement:** Not applicable.

**Conflicts of Interest:** The authors declare no conflict of interest.

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
