# Peer review of "On Advances of Lattice-Based Cryptographic Schemes and Their Implementations"

_cryptography, doi:10.3390/cryptography6040056_

Round 1

Reviewer 1 Report

See report.

Author Response

Issues are addressed in the manuscript and mentioned them in the response report.

Reviewer 2 Report

·       The topics in the introduction section, such as RSA algorithms, Diffie-Helman, post-quantum cryptographic schemes,etc., should be moved to separate sections, such as background, because the inclusion of these topics makes the introduction difficult to understand.

·       Paper organisation paragraph is missing in the paper.

·       A detailed comparison is needed in each section. A survey paper should not only cover the introduction of methods.

·       A discussion section is needed on what the authors have learned in this survey and what new information has been derived by performing this survey.  

·       The conclusion section should be more precise and in summarised form.

·       There are many typo errors existed in the paper, such as

o   In keywords, Implementation)

o   In section 5, Kyber[20], NewHope[23]

·       The latest references (2021,2022, ) are completely missing

Author Response

(The authors gave the same response as above.)

Reviewer 3 Report

The paper is almost comprehensive, however most recent references from 2021 upwards are missing, which is essential in a survey. Paper has several typos, e.g. in lines 129, 331, 430, 443, 728.

Author Response

(The authors gave the same response as above.)

Round 2

Reviewer 1 Report

The authors have implemented my many minor corrections and suggestions, and have followed a few of the more general suggestions as well. This had improved the presentation of the paper considerably.
I am still not sure that the paper is suitable for publication - it is very weak and unoriginal, still verging on plagiarism - but it is possibly useful as a quick collection of brief descriptions of recent methods. For this reason, I tentatively recommend the paper for publication. The authors should try to improve the language further - there are still many small errors of grammar, for instance, and vague sentences.

Author Response

Thank you very much for your comments. Response is attached.

Reviewer 2 Report

The authors have not addressed the comments properly, as suggested. 

For example, a detailed comparison is still missing, as highlighted before

Further, the introduction section only consists of one paragraph.

The discussion section is not provided, but it was requested.

Author Response

(The authors gave the same response as above.)

Reviewer 3 Report

Please, check the following lines for minor improvements or typos: 89, 130 (the order function I guess it is wrong), 180, 247, 

Author Response

(The authors gave the same response as above.)

Round 3

Reviewer 2 Report

The authors have revised the suggested changes.